# An Error Detection and Correction Framework for Connectomics

**Jonathan Zung**
Princeton University
`jzung@princeton.edu`

**Ignacio Tartavull**
Princeton University
`tartavull@princeton.edu`

**Kisuk Lee**
Princeton University and MIT
`kisuklee@mit.edu`

**H. Sebastian Seung**
Princeton University
`sseung@princeton.edu`

## Abstract

We define and study error detection and correction tasks that are useful for 3D reconstruction of neurons from electron microscopic imagery, and for image segmentation more generally. Both tasks take as input the raw image and a binary mask representing a candidate object. For the error detection task, the desired output is a map of split and merge errors in the object. For the error correction task, the desired output is the true object. We call this object mask pruning, because the candidate object mask is assumed to be a superset of the true object. We train multiscale 3D convolutional networks to perform both tasks. We find that the error-detecting net can achieve high accuracy. The accuracy of the error-correcting net is enhanced if its input object mask is "advice" (union of erroneous objects) from the error-detecting net.

## 1 Introduction

While neuronal circuits can be reconstructed from volumetric electron microscopic imagery, the process has historically [30] and even recently [28] been highly laborious. One of the most time-consuming reconstruction tasks is the tracing of the brain's "wires," or neuronal branches. This task is an example of instance segmentation, and can be automated through computer detection of the boundaries between neurons. Convolutional nets were first applied to neuronal boundary detection a decade ago [10, 29]. Since then convolutional nets have become the standard approach, and the accuracy of boundary detection has become impressively high [31, 1, 15, 6].

Given the low error rates, it becomes helpful to think of subsequent processing steps in terms of modules that detect and correct errors. In the error detection task (Figure 1a), the input is the raw image and a binary mask that represents a candidate object. The desired output is a map containing the locations of split and merge errors in the candidate object. Related work on this problem has been restricted to detection of merge errors only by either hand-designed [18] or learned [24] computations. However, a typical segmentation contains both split and merge errors, so it would be desirable to include both in the error detection task.

In the error correction task (Figure 1b), the input is again the raw image and a binary mask that represents a candidate object. The candidate object mask is assumed to be a superset of a true object, which is the desired output. With this assumption, error correction is formulated as *object mask pruning*. Object mask pruning can be regarded as the splitting of undersegmented objects to create true objects. In this sense, it is the opposite of agglomeration, which merges oversegmented objects to create true objects [11, 21]. Object mask pruning can also be viewed as the *subtraction* of voxels

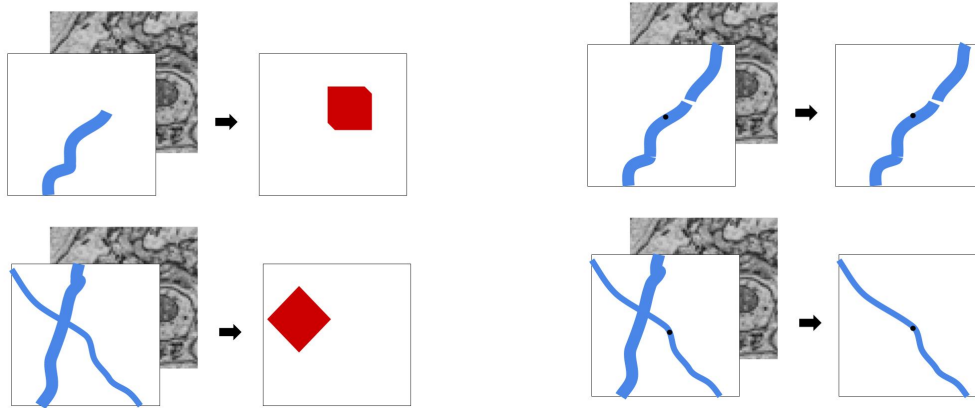

(a) Error detection task for split (top) and merge (bottom) errors. The desired output is an error map (red). A voxel in the error map is red if and only if a window centered on it contains a split or merge error. We also consider a variant of the task in which the object mask is the sole input; the grayscale image is not used.

(b) The object mask pruning task. The input mask is assumed to be a superset of a true object. The desired output (right) is the true object containing the central voxel (black dot). In the first case there is nothing to prune, while in the second case the object not overlapping the central voxel is erased.

Figure 1: Error detection and correction tasks. For both tasks, the inputs are a candidate object mask (blue) and the original image (grayscale). Note that diagrams are 2D for illustrative purposes, but in reality the inputs and outputs are 3D.

from an object to create a true object. In this sense, it is the opposite of a flood-filling net [13, 12] or *MaskExtend* [18], each iteration of which is the *addition* of voxels to an object to create a true object. Iterative mask extension has been studied in other work on instance segmentation in computer vision [25, 23]. The task of generating an object mask *de novo* from an image has also been studied in computer vision [22].

We implement both error detection and error correction using 3D multiscale convolutional networks. One can imagine multiple uses for these nets in a connectomics pipeline. For example, the error-detecting net could be used to reduce the amount of labor required for proofreading by directing human attention to locations in the image where errors are likely. This labor reduction could be substantial because the declining error rate of automated segmentation has made it more time-consuming for a human to find an error.

We show that the error-detecting net can provide "advice" to the error-correcting net in the following way. To create the candidate object mask for the error-correcting net from a baseline segmentation, one can simply take the union of all erroneous segments as found by the error-detecting net. Since the error rate in the baseline segmentation is already low, this union is small and it is easy to select out a single object. The idea of using the error detector to choose locations for the error corrector was proposed previously though not actually implemented [18]. Furthermore, the idea of using the error detector to not only choose locations but provide "advice" is novel as far as we know.

We contend that our approach decomposes the neuron segmentation problem into two strictly easier pieces. First, we hypothesize that recognizing an error is much easier than producing the correct answer. Indeed, humans are often able to detect errors using only morphological cues such as abrupt terminations of axons, but may have difficulty actually finding the correct extension.

On the other hand, if the error-detection network has high accuracy and the initial set of errors is sparse, then the error correction module only needs to prune away a small number of irrelevant parts from the candidate mask described above. This contrasts with the flood-filling task which involves an unconstrained search for new parts to add. Given that most voxels are *not* a part of the object to be reconstructed, an upper bound on the object is usually more informative than a lower bound. As an added benefit, selective application of the error correction module near likely errors makes efficient use of our computational budget [18].

In this paper, we support the intuition above by demonstrating high accuracy detection of both split and merge errors. We also demonstrate a complete implementation of the stated error detection-correction framework, and report significant improvements upon our baseline segmentation.

Some of the design choices we made in our neural networks may be of interest to other researchers. Our error-correcting net is trained to produce a vector field via metric learning instead of directly producing an object mask. The vector field resembles a semantic labeling of the image, so this approach blurs the distinction between instance and semantic segmentation. This idea is relatively new in computer vision [7, 4, 3]. Our multiscale convolutional net architecture, while similar in spirit to the popular U-Net [26], has some novelty. With proper weight sharing, our model can be viewed as a feedback recurrent convolutional net unrolled in time (see the appendix for details). Although our model architecture is closely related to the independent works of [27, 9, 5], we contribute a *feedback recurrent* convolutional net interpretation.

## 2    Error detection

### 2.1    Task specification: detecting split and merge errors

Given a single segment in a proposed segmentation presented as an object mask $Obj$, the error detection task is to produce a binary image called the *error map*, denoted $Err_{p_x \times p_y \times p_z}(Obj)$. The definition of the error map depends on a choice of a window size $p_x \times p_y \times p_z$. A voxel $i$ in the error map is 0 if and only if the restriction of the input mask to a window centred at $i$ of size $p_x \times p_y \times p_z$ is voxel-wise equal to the restriction of some object in the ground truth. Observe that the error map is sensitive to both split and merge errors.

A smaller window size allows us to localize errors more precisely. On the other hand, if the window radius is less than the width of a typical boundary between objects, it is possible that two objects participating in a merge error never appear in the same window. These merge errors would not be classified as an error in any window.

We could use a less stringent measure than voxel-wise equality that disregards small perturbations of the boundaries of objects. However, our proposed segmentations are all composed of the same building blocks (supervoxels) as the ground truth segmentation, so this is not an issue for us.

We define the *combined error map* as $\sum_{Obj} Err(Obj) * Obj$ where $*$ represents pointwise multiplication. In other words, we restrict the error map for each object to the object itself, and then sum the results. The figures in this paper show the *combined error map*.

### 2.2    Architecture of the error-detecting net

We take a fully supervised approach to error detection. We implement error detection using a multiscale 3D convolutional network. The architecture is detailed in Figure 2. Its design is informed by experience with convolutional networks for neuronal boundary detection (see [15]) and reflects recent trends in neural network design [26, 8]. Its field of view is $P_x \times P_y \times P_z = 318 \times 318 \times 33$ (which is roughly cubic in physical size given the anisotropic resolution of our dataset). The network computes (a downsampling of) $Err_{46 \times 46 \times 7}$. At test time, we perform inference in overlapping windows and conservatively blend the output from overlapping windows using a maximum operation.

We trained two variants, one of which takes as input only $Obj$, and another which additionally receives as input the raw image.

## 3    Error correction

### 3.1    Task specification: object mask pruning

Given an image patch of size $P_x \times P_y \times P_z$ and a candidate object mask of the same dimensions, the *object mask pruning* task is to erase all voxels which do not belong to the true object overlapping the central voxel. The candidate object mask is assumed to be a superset of the true object.

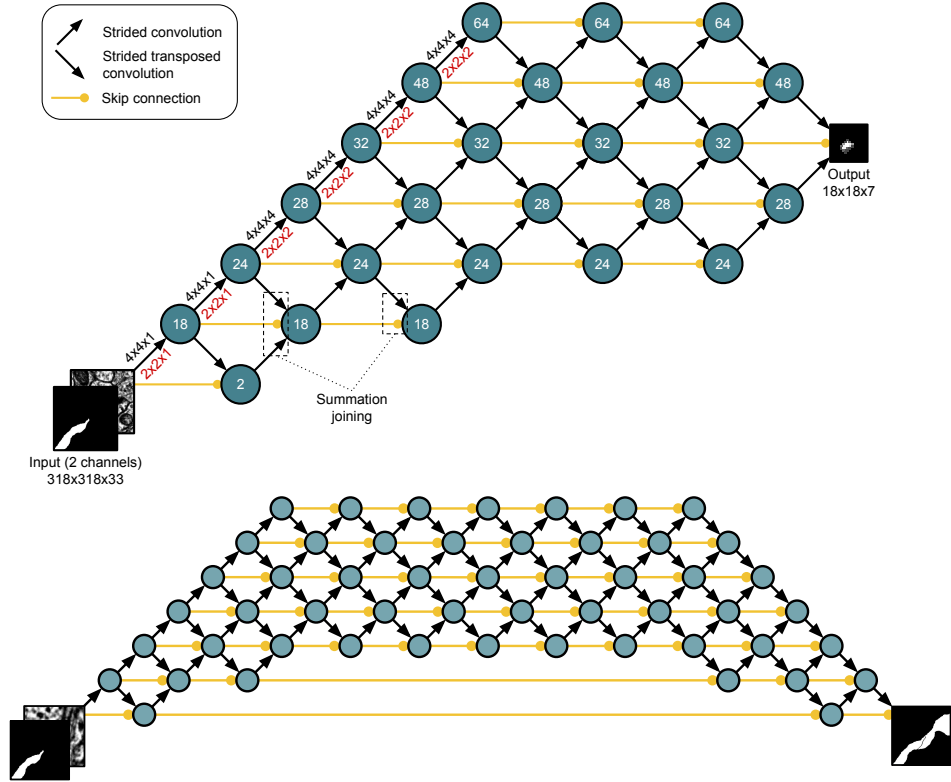

Figure 2: Architectures for the error-detecting and error-correcting nets respectively. Each node represents a layer and the number inside represents the number of feature maps. The layers closer to the top of the diagram have lower resolution than the layers near the bottom. We make savings in computation by minimizing the number of high resolution feature maps. The diagonal arrows represent strided convolutions, while the horizontal arrows represent skip connections. Associated with the diagonal arrows, black numbers indicate filter size and red numbers indicate strides in $x \times y \times z$. Due to the anisotropy of the resolution of the images in our dataset, we design our nets so that the first convolutions are exclusively 2D while later convolutions are 3D. The field of view of a unit in the higher layers is therefore roughly cubic. To limit the number of parameters in our model, we factorize all 3D convolutions into a 2D convolution followed by a 1D convolution in $z$-dimension. We also use weight sharing between some convolutions at the same height. Note that the error-correcting net is a prolonged, symmetric version of the error-detecting net. For more detail of the error corrector, see the appendix.

## 3.2 Architecture of the error-correcting net

Yet again, we implement error correction using a multiscale 3D convolutional network. The architecture is detailed in Figure 2. One difficulty with training a neural network to reconstruct the object containing the central voxel is that the desired output can change drastically as the central voxel moves between objects. We use an intermediate representation whose role is to soften this dependence on the location of the central voxel. The desired intermediate representation is a $k = 6$ dimensional vector $v(x, y, z)$ at each point $(x, y, z)$ such that points within the same object have similar vectors and points in different objects have different vectors. We transform this vector field into a binary image $M$ representing the object overlapping the central voxel as follows:

$$M(x, y, z) = \exp\left(-||v(x, y, z) - v(0, 0, 0)||^2\right),$$

where $(0, 0, 0)$ is the central voxel. When an over-segmentation is available, we replace $v(0, 0, 0)$ with the average of $v$ over the supervoxel containing the central voxel. This trick makes it unnecessary to centre our windows far away from a boundary, as was necessary in [13]. Note that we backpropagate

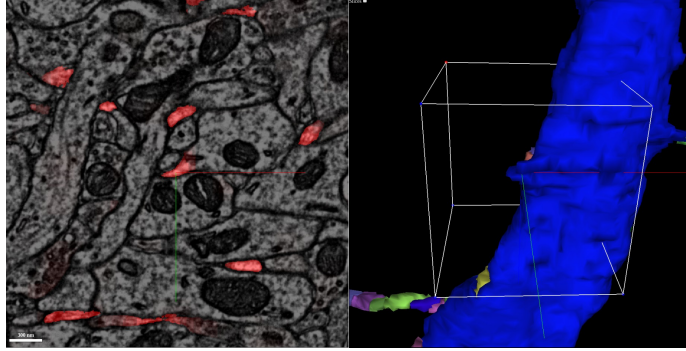

Figure 3: An example of a mistake in the initial segmentation. The dendrite is missing a spine. The red overlay on the left shows the combined error map (defined in Section 2.1); the stump in the centre of the image was clearly marked as an error.

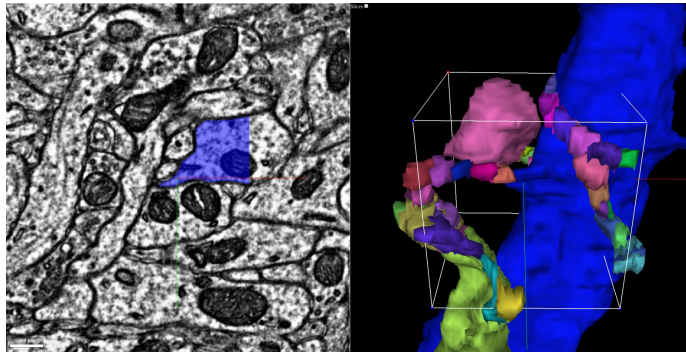

Figure 4: The right shows all objects which contained a detected error in the vicinity. For clarity, each supervoxel was drawn with a different colour. The union of these objects is the binary mask which is provided as input to the error correction network. For clarity, these objects were clipped to lie within the white box representing the field of view of our error correction network. The output of the error correction network is overlaid in blue on the left.

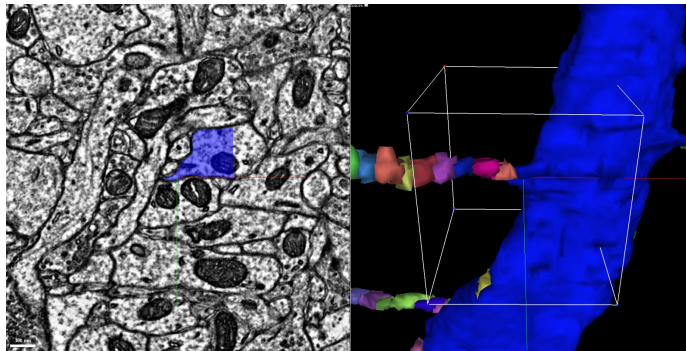

Figure 5: The supervoxels assembled in accordance with the output of the error correction network.

through the transform $M$, so the vector representation may be seen as an implementation detail and the final output of the network is just a (soft) binary image.

## 4 How the error detector can "advise" the error corrector

Suppose that we would like to correct the errors in a baseline segmentation. Obviously, the error-detecting net can be used to find locations where the error-correcting net can be applied [18]. Less obviously, the error-detecting net can be used to construct the object mask that is the input to the error-correcting net. We refer to this object mask as the "advice mask" and its construction is

important because the baseline object to be corrected might contain split as well as merge errors, while the object mask pruning task can correct only merge errors.

The advice mask is defined is the union of the baseline object at the central pixel with all other baseline objects in the window that contain errors as judged by the error-detecting net. The advice mask is a superset of the true object overlapping the central voxel, assuming that the error-detecting net makes no mistakes. Therefore advice is suitable as an input to the object mask pruning task.

The details of the above procedure are as follows. We begin with an initial baseline segmentation whose remaining errors are assumed to be sparsely distributed. During the error correction phase, we iteratively update a segmentation represented as the connected components of a graph $G$ whose vertices are segments in a strict over-segmentation (henceforth called supervoxels). We also maintain the combined error map associated with the current segmentation. We binarize the error map by thresholding it at 0.25.

Now we iteratively choose a location $\ell = (x, y, z)$ which has value 1 in the binarized combined error map. In a $P_x \times P_y \times P_z$ window centred on $\ell$, we prepare an input for the error corrector by taking the union of all segments containing at least one white voxel in the error map. The error correction network produces from this input a binary image $M$ representing the object containing the central voxel. For each supervoxel $S$ touching the central $P_x/2 \times P_y/2 \times P_z/2$ window, let $M(S)$ denote the average value of $M$ inside $S$. If $M(S) \notin [0.1, 0.9]$ for all $S$ in the relevant window (i.e. the error corrector is confident in its prediction for each supervoxel), we add to $G$ a clique on $\{S \mid M(S) > 0.9\}$ and delete from $G$ all edges between $\{S \mid M(S) < 0.1\}$ and $\{S \mid M(S) > 0.9\}$. The effect of these updates is to change $G$ to locally agree with $M$. Finally, we update the combined error map by applying the error detector at all locations where its decision could have changed.

We iterate until every location is zero in the error map or has been covered by a window at least $t = 2$ times by the error corrector. This stopping criterion guarantees that the algorithm terminates. In practice, the segmentation converges without this auxiliary stopping condition to a state in which the error corrector fails confidence threshold everywhere. However, it is hard to certify convergence since it is possible that the error corrector could give different outputs on slightly shifted windows. Based on our validation set, increasing $t$ beyond 2 did not measurably improve performance.

Note that this algorithm deals with split and merge errors, but cannot fix errors already present at the supervoxel level.

# 5 Experiments

## 5.1 Dataset

Our dataset is a sample of mouse primary visual cortex (V1) acquired using serial section transmission electron microscopy at the Allen Institute for Brain Science. The voxel resolution is $3.6\,\text{nm} \times 3.6\,\text{nm} \times 40\,\text{nm}$.

Human experts used the VAST software tool [14, 2] to densely reconstruct multiple volumes that amounted to 530 Mvoxels of ground truth annotation. These volumes were used to train a neuronal boundary detection network (see the appendix for architecture). We applied the resulting boundary detector to a larger volume of size 4800 Mvoxels to produce a preliminary segmentation, which was then proofread by the tracers. This bootstrapped ground truth was used to train the error detector and corrector. A subvolume of size 910 Mvoxels was reserved for validation, and a subvolume of size 910 Mvoxels was reserved for testing.

Producing the gold standard segmentation required a total of $\sim 560$ tracer hours, while producing the bootstrapped ground truth required $\sim 670$ tracer hours.

## 5.2 Baseline segmentation

Our baseline segmentation was produced using a pipeline of multiscale convolutional networks for neuronal boundary detection, watershed, and mean affinity agglomeration [15]. We describe the pipeline in detail in the appendix. The segmentation performance values reported for the baseline are taken at a mean affinity agglomeration threshold of 0.23, which minimizes the variation of information error metric [17, 20] on the test volumes.

### 5.3 Training procedures

**Sampling procedure**   Here we describe our procedure for choosing a random point location in a segmentation. Uniformly random sampling is unsatisfactory since large objects such as dendritic shafts will be overrepresented. Instead, given a segmentation, we sample a location $(x, y, z)$ with probability inversely proportional to the fraction of a window of size $128 \times 128 \times 16$ centred at $(x, y, z)$ which is occupied by the object containing the central voxel.

**Training of error detector**   An initial segmentation containing errors was produced using our baseline neuronal boundary detector combined with mean affinity agglomeration at a threshold of 0.3. Point locations were sampled according to the sampling procedure specified in 5.3. We augmented all of our data with rotations and reflections. We used a pixelwise cross-entropy loss.

**Training of error corrector**   We sampled locations in the ground truth segmentation as in 5.3. At each location $\ell = (x, y, z)$, we generated a training example as follows. Let $Obj_\ell$ be the ground truth object touching $\ell$. We selected a random subset of the objects in the window centred on $\ell$ including $Obj_\ell$. To be specific, we chose a number $p$ uniformly at random from $[0, 1]$, and then selected each segment in the window with probability $p$ in addition $Obj_\ell$. The input at $\ell$ was then a binary mask representing the union of the selected objects along with the raw EM image, and the desired output was a binary mask representing only $Obj_\ell$. The dataset was augmented with rotations, reflections, simulated misalignments and missing sections [15]. We used a pixelwise cross-entropy loss.

Note that this training procedure uses only the ground truth segmentation and is completely independent of the error detector and the baseline segmentation. This convenient property is justified by the fact that if the error detector is perfect, the error corrector only ever receives as input unions of complete objects.

### 5.4 Error detection results

To measure the quality of error detection, we densely sampled points in our test volume as in 5.3. In order to remove ambiguity over the precise location of errors, we filtered out points which contained an error within a surrounding window of size $80 \times 80 \times 8$ but not a window of size $40 \times 40 \times 4$. These locations were all unique, in that two locations in the same object were separated by at least $80, 80, 8$ in $x, y, z$, respectively. Precision and recall simultaneously exceed 90% (see Figure 6). Empirically, many of the false positive examples come where a dendritic spine head curls back and touches its trunk. These examples locally appear to be incorrectly merged objects.

We trained one error detector with access to the raw image and one without. The network's admirable performance even without access to the image as seen in Figure 6 supports our hypothesis that error detection is a relatively easy task and can be performed using only shape cues.

Merge errors qualitatively appear to be especially easy for the network to detect; an example is shown in Figure 7.

### 5.5 Error correction results

Table 1: Comparing segmentation performance

|                | $VI_{\text{merge}}$ | $VI_{\text{split}}$ | Rand Recall | Rand Precision |
|----------------|---------------------|---------------------|-------------|----------------|
| Baseline       | 0.162               | 0.142               | 0.952       | 0.954          |
| Without Advice | 0.130               | 0.057               | 0.956       | 0.979          |
| With Advice    | **0.088**           | **0.052**           | **0.974**   | **0.980**      |

In order to demonstrate the importance of error detection to error correction, we ran two experiments: one in which the binary mask input to the error corrector was simply the union of all segments in the window ("without advice"), and one in which the binary mask was the union of all segments with a detected error ("with advice"). In the "without advice" mode, the network is essentially asked to reconstruct the object overlapping the central voxel in one shot. Table 1 shows that advice confers a considerable advantage in performance on the error corrector.

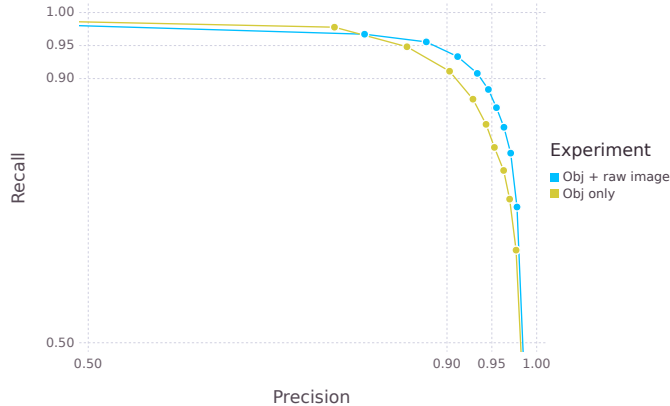

Figure 6: Precision and recall for error detection, both with and without access to the raw image. In the test volume, there are $8248$ error free locations and $944$ locations with errors. In practice, we use threshold which guarantees $> 95\%$ recall and $> 85\%$ precision.

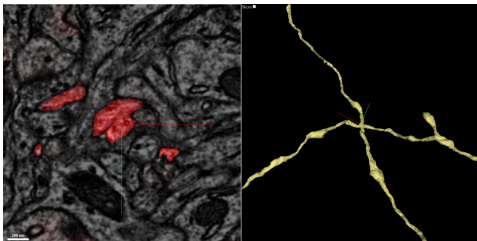

Figure 7: An example of a detected error. The right shows two incorrectly merged axons, and the left shows the predicted combined error map (defined in 2.1) overlaid on the corresponding 2D image in red.

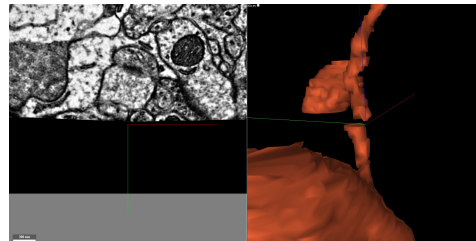

Figure 8: A difficult location with missing data in one section combined with a misalignment between sections. The error-correcting net was able to trace across the missing data.

It is sometimes difficult to assess the significance of an improvement in variation of information or rand score since changes can be dominated by modifications to a few large objects. Therefore, we decomposed the variation of information into a score for each object in the ground truth. Figure 9 summarizes the cumulative distribution of the values of $VI(i) = VI_{merge}(i) + VI_{split}(i)$ for all segments $i$ in the ground truth. See the appendix for a precise definition of $VI(i)$.

The number of errors from the set in Sec. $5.4$ that were fixed or introduced by our iterative refinement procedure is shown in 2. These numbers should be taken with a grain of salt since topologically insignificant changes could count as errors. Regardless, it is clear that our iterative refinement procedure fixed a significant fraction of the remaining errors and that "advice" improves the error corrector.

The results are qualitatively impressive as well. The error-correcting network is sometimes able to correctly merge disconnected objects, for example in Figure 8.

Table 2: Number of errors fixed and introduced relative to the baseline

|  | # Errors | # Errors fixed | # Errors introduced |
|---|---|---|---|
| Baseline | 944 | - | - |
| Without Advice | 474 | 547 | 77 |
| With Advice | **305** | 707 | 68 |

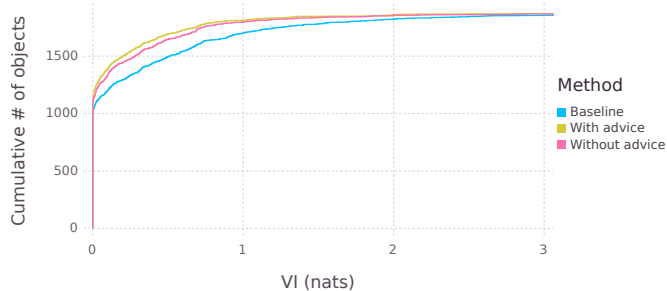

Figure 9: Per-object VI scores for the 940 reconstructed objects in our test volume. Almost 800 objects are completely error free in our segmentation. These objects are likely all axons; almost every dendrite is missing a few spines.

## 5.6 Computational cost analysis

Table 3 shows the computational cost of the most expensive parts of our segmentation pipeline. Boundary detection and error detection are run on the entire image, while error correction is run on roughly 10% of the possible locations in the image. Error correction is still the most costly step, but it would be $10\times$ more costly without restricting to the locations found by the error detection network. Therefore, the cost of error detection is more than justified by the subsequent savings during the error correction phase. The number of locations requiring error correction will fall even further if the precision of the error detector increases or the error rate of the initial segmentation decreases.

Table 3: Computation time for a $2048 \times 2048 \times 256$ volume using a single TitanX Pascal GPU

| | |
|---|---|
| Boundary Detection | 18 mins |
| Error Detection | 25 mins |
| Error Correction | 55 mins |

## 6 Conclusion and future directions

We have developed a error detector for the neuronal segmentation problem and combined it with an error correction module. In particular, we have shown that our error detectors are able to exploit priors on neuron shape, having reasonable performance even without access to the raw image. We have made significant savings in computation by applying expensive error correction procedures only where predicted necessary by the error detector. Finally, we have demonstrated that the "advice" of error detection improves an error correction module, improving segmentation performance upon our baseline.

We expect that significant improvements in the accuracy of error detection could come from aggressive data augmentation. We can mutilate a ground truth segmentation in arbitrary (or even adversarial) ways to produce unlimited examples of errors.

An error detection module has many potential uses beyond the ones presented here. For example, we could use error detection to direct ground truth annotation effort toward mistakes. If sufficiently accurate, it could also be used directly as a learning signal for segmentation algorithms on unlabelled data. The idea of co-training our error-correction and error-detection networks is natural in view of recent work on generative adversarial networks [19, 16].

**Author contributions and acknowledgements**

JZ conceptualized the study and conducted most of the experiments and evaluation. IT (along with Will Silversmith) created much of the infrastructure necessary for visualization and running our algorithms at scale. KL produced the baseline segmentation. HSS helped with the writing.

We are grateful to Clay Reid, Nuno da Costa, Agnes Bodor, Adam Bleckert, Dan Bumbarger, Derrick Britain, JoAnn Buchannan, and Marc Takeno for acquiring the TEM dataset at the Allen Institute for Brain Science. The ground truth annotation was created by Ben Silverman, Merlin Moore, Sarah Morejohn, Selden Koolman, Ryan Willie, Kyle Willie, and Harrison MacGowan. We thank Nico Kemnitz for proofreading a draft of this paper. We thank Jeremy Maitin-Shepard at Google and the other contributors to the neuroglancer project for creating an invaluable visualization tool.

We acknowledge NVIDIA Corporation for providing us with early access to Titan X Pascal GPU used in this research, and Amazon for assistance through an AWS Research Grant. This research was supported by the Mathers Foundation, the Samsung Scholarship and the Intelligence Advanced Research Projects Activity (IARPA) via Department of Interior/ Interior Business Center (DoI/IBC) contract number D16PC0005. The U.S. Government is authorized to reproduce and distribute reprints for Governmental purposes notwithstanding any copyright annotation thereon. Disclaimer: The views and conclusions contained herein are those of the authors and should not be interpreted as necessarily representing the official policies or endorsements, either expressed or implied, of IARPA, DoI/IBC, or the U.S. Government.

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
