[Supplementary Material]

# Supplementary Information for "An Error Detection and Correction Framework for Connectomics" by Zung et al.

## A    Baseline Neuronal Boundary Detection

In this section, we describe our baseline segmentation pipeline, which is similar to what is described in [8]. The major difference is our novel densely multiscale 3D convolutional network architecture for neuronal boundary detection, which will be described in detail in the following section. (The same *class* of architecture was employed in error detection and error correction. See main text.)

### A.1    Network architecture

Our proposed densely multiscale 3D convolutional network for neuronal boundary detection is illustrated in Figure 1. Our model is built upon U-Net [12] with several interesting architectural augmentation. Our model can be viewed as a pyramidal stack of the basic computational module (diamond-shaped box in Figure 1). This diamond-shaped module can be interpreted as a residual building block (see Figure 2 in [5]) with two residual pathways, one top-down and the other bottom-up. Thus our model is *fully residual* in the sense that every computational pathway involving horizontal information flow is passing through the residual module. Moreover, every residual module refines its input representation by integrating both top-down and bottom-up information, thus allowing for *dense* intermixing of multiscale features. Our model's *dense* and *fully residual* architecture allows an incremental and iterative top-down/bottom-up refinement of internal representation, which is in contrast to U-Net and variants' more restricted coarse-to-fine top-down refinement [11, 9].

From a different point of view, Figure 2 illustrates another important motivation for our densely multiscale convolutional net architecture. Our model can be viewed as a feedback recurrent convolutional network unrolled in time (Figure 2). Weight-sharing across time makes a our model exactly equivalent to a convolutional net with recurrent feedback connections unfolded through time, and this novel perspective provides a better framework for understanding one of the unique characteristics

Figure 1: Architecture for the baseline neuronal boundary detection. Each node represents a layer and the number inside represents the number of feature maps. The layers closer to the top of the diagram have lower resolution than the layers near the bottom. The diagonal arrows represent strided convolutions, while the horizontal arrows represent skip connections. Associated with the diagonal arrows, black numbers indicate filter size and red numbers indicate strides in $x \times y \times z$. The target for our boundary detection net is a 3D *affinity graph* [13, 8, 3], thus outputting three channels of each corresponding to $x$ (green), $y$ (red), and $z$ (blue) affinity map, respectively.

Figure 2: Feedback recurrent convolutional network unrolled in time. See the text in Section A.1 for further details.

of our model, i.e., the incremental refinement of internal representation by interative integration of top-down and bottom-up information. Our net's internal representation is incremetally and iteratively refined over time by integrating the top-down contextual information conveyed through the feedback recurrent connnections and the higer spatial-frequency information relayed through the bottom-up feedforward connections.

### A.1.1 Architectural details

Due to the anisotropy of the resolution of the images in our dataset, we design our networks such that the first convolutions are exclusively 2D while later convolutions are 3D (see Figure 1). The field of view of a unit in the higher layers is therefore roughly cubic. To limit the number of parameters in our model, we factorize all 3D convolutions into a 2D convolution followed by a 1D convolution in $z$-dimension. We employ exponential linear units (ELUs, [2]) for nonlinearity, except for the output layer with logistic activation functions.

### A.2 Dataset

Our dataset is a sample of mouse primary visual cortex (V1) acquired using transmission electron microscopy at the Allen Institute for Brain Science. The voxel resolution is $3.6 \text{ nm} \times 3.6 \text{ nm} \times 40 \text{ nm}$.

A team of tracers produced multiple volumes of gold standard dense reconstruction, in total 20 volumes of size $512 \times 512 \times 100$. We trained our boundary detector using 19 volumes and used the last volume for training validation. We then applied the trained boundary detector on a new image volume of size $2048 \times 2048 \times 100$ to obtain a preliminary segmentation, which was then proofread by the tracers to generate a bootstrapped ground truth volume. This volume was used to optimize the parameters for watershed and mean affinity agglomeration. Finally, the optimized segmentation pipeline was applied to generate further bootstrapped ground truth for the error detection and correction.

### A.3 Training procedures

Our boundary detection networks were implemented based on the Caffe deep learning framework [6]. To train our models, we minimized the binomial cross-entropy loss with class-rebalancing using the Adam optimizer [7], initialized with $\alpha = 0.001$, $\beta_1 = 0.9$, $\beta_2 = 0.999$, and $\epsilon = 0.01$. The network weights were initialized following He et al. [4]. The learning rate (or step size parameter $\alpha$ in the Adam optimizer) was halved when validation loss plateaued out, five times in total at 35K, 175K, 250K, 300K, and 480K training iterations. We used a single patch of size $158 \times 158 \times 32$ (i.e. minibatch of size 1) to compute gradients at each training iteration. The training lasted for 800K iterations until convergence, which took about five days on a single NVIDIA Titan X Pascal GPU.

### A.4 Inference and postprocessing

We perform *overlap-blending* inference followed by watershed and mean affinity agglomeration [8]. We refer the interested readers to [8] for further details.

## B  Per-object VI score

Recall that the variation of information between two segmentations may be computed as

$$VI_{\text{split}} = -\frac{1}{\sum_{i,j} r_{ij}} \sum_{i,j} r_{ij} \log(r_{ij}/p_i),$$

$$VI_{\text{merge}} = -\frac{1}{\sum_{i,j} r_{ij}} \sum_{i,j} r_{ij} \log(r_{ij}/q_j),$$

$$p_i = \sum_j r_{ij},$$

$$q_j = \sum_i r_{ij},$$

where $r_{ij}$ is the number of voxels in common between the $i^{\text{th}}$ segment of the ground truth segmentation and the $j^{\text{th}}$ segment of the proposed segmentation [10].

We define the split and merge scores for ground truth segment $i$ as

$$VI_{\text{split}}(i) = -\sum_j r_{ij}/p_i \log(r_{ij}/p_i),$$

$$VI_{\text{merge}}(i) = -\sum_j r_{ij}/p_i \log(r_{ij}/q_j),$$

Both quantities have units of bits. $VI_{\text{split}}(i)$ is zero iff ground truth segment $i$ is contained within a segment in the proposed segmentation, while $VI_{\text{merge}}(i)$ is zero iff ground truth segment $i$ is the union of one or more segments in the proposed segmentation. The total score $VI_{\text{split}}$ or $VI_{\text{merge}}$ is a weighted sum of the per-object scores $VI_{\text{split}}(i)$, $VI_{\text{merge}}(i)$ respectively.

## C  Training details

The error-detecting and error-correcting networks were implemented in TensorFlow [1] and trained using 4 TitanX Pascal GPUs with synchronous gradient descent. We used the Adam optimizer, initialized with $\alpha = 0.001$, $\beta_1 = 0.95$, $\beta_2 = 0.9995$, and $\epsilon = 0.1$ [7]. Both nets were trained until the loss on a validation set plateaued. The error-detecting net was trained for 700K iterations (approximately one week), while the error-correcting net was trained for 1.7M iterations (approximately three weeks).

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

Figure 3: Error-correcting network architecture