[Reviews · NeurIPS 2017]

Reviewer 1



The paper propsoes two algorithms, one for finding and one for correcting errors in the shape of segments of electron microscopy images for neural circuit reconstruction. It combines these algorithms in a heuristic fashion so as to greedily update an initial segmentation. * Contribution I understand that the first deep network (called the detector) estimates, for any ROI, whether the segment overlapping the central pixel in this ROI needs correction. I also understand that the second deep network (called the corrector) estimates the shape of the correct segment that overlaps with the central pixel. It seems to me that, taking only the output of the corrector, at every pixel, and a metric of segmentations such as the VI, one obtains a consensus problem for the entire volume. However, this problem is not studied or even mentioned in this paper. Instead, some greedy algorithm (described informally in Section 5) based in some way on some superpixel segmentation (mentioned in Section 5) is used to incrementally update an initial segmentation. While the empirical results may be of interest to a connectomics audience (which I cannot judge), I doubt that this greedy algorithm has applications beyong connectomics. I conclude that this paper is not of sufficient interest to the wider NIPS audience. * Presentation The paper is well-structured and quite well-written, except for the technical sections 3, 4 and especially 5 that are entirely informal, which makes it hard to understand how the algorithms work in exact detail. The relation of a problem that is hard to solve to a problem whose solution is easy to verify should be toned down in the introduction as well as in the discussion of related work, as this idea is familiar to most computer scientists from the definition of the complexity class NP. In particular, I do not agree "that the error detection task is better posed than the supervoxel agglomeration task." * Related work The relation of the proposed method to GANs [8,9] and visual attention [10] is far-fetched, speculative and not supported in this work by technical arguments. If the authors wish to hint on such connections, they should do so only in an outlook, at the end of the paper.

Reviewer 2



A nice targeted error detection and correction framework for connectomics, leading to well quantified improvements in performance.

Reviewer 3



The paper tackles the important, but relatively unexplored, area of error detection and targeted error correction in connectomics. There is a clear need for this type of methods in the field, primarily in order to optimize the allocation of human proofreading time. The authors make the observation that to a human segmentation errors are much easier to detect than to correct, and propose to automate this process. Specifically, they propose to build an error detector module in the form a multi-scale 3d CNN, taking as input a binary object mask and predicting whether it is equal to an object in the ground truth segmentation. At inference time, the network is applied on overlapping windows distributed over a grid to identify and localize errors in the segmentation. The paper also proposes an error correction module -- a 3d CNN reconstructing the object containing the central pixel, similarly to flood-filling networks (FFNs), which the authors cite as related work. Instead of predicting a binary mask like in FFNs, The authors propose to predict a k-dimensional vector for each point of the output, so voxels of the same object have a similar vector, and different objects have not. This vector field is then transformed into a binary mask with an exponential transform. The stated goal of this approach is to soften the dependency on the precise location of the central object. The authors should consider including some additional information about why they chose this particular form of the transform to generate the binary mask, and whether other forms were considered; what value of k was used in the experiments, as well as any experimental data showing that this approach indeed improves the results compared to a direct binary encoding of the output. The paper is missing information on which losses were used to train the network, which seems particularly important for the error correction module where the k-dimensional vectors are arbitrary and presumably cannot be fully specified based on the training data. In section 5, the confidence threshold should be explained in more detail -- how was this confidence computed, and what threshold was used? The text also states that the termination condition was an error-free state as predicted by the error detector, or two corrections per iteration. Why was this particular condition used? Would applying the network more than two times result in a better segmentation? Would such a process converge to a stationary state of the segmentation? It should also be explained explicitly, whether the error corrector deals with both split and merge errors, or only split errors. If merge errors are handled, it should be mentioned what happens in case they are at the supervoxel graph level, and within the supervoxels itself. In section 6.3, the parameters of the Adam optimizer should be explicitly mentioned. The paper presents a detailed analysis of performance of the presented systems. The authors should be commended for computing per-object VI scores to make the metrics easier to understand. Technical comments about the text: - Fig. 1 is hard to understand, and it is unclear what the nodes represent. The caption mentions horizontal layers, but it does not look like the data flows through the network vertically. - Line 48 states that the "overhead of agglomeration prohibits the use of deep learning". This does not seem to be the case, as deep nets have been used for agglomeration, see e.g. https://papers.nips.cc/paper/6595-combinatorial-energy-learning-for-image-segmentation.pdf for a recent example. - Appendix A, Table 4: Why are layer IDs not consecutive, and if this is intentional, what do the IDs mean?